# The Synergistic Effect of Piperlongumine and Sanguinarine on the Non-Small Lung Cancer

**DOI:** 10.3390/molecules25133045

**Published:** 2020-07-03

**Authors:** Marta Hałas-Wiśniewska, Wioletta Zielińska, Magdalena Izdebska, Alina Grzanka

**Affiliations:** Department of Histology and Embryology, Faculty of Medicine, Nicolaus Copernicus University in Toruń, Collegium Medicum in Bydgoszcz, Karłowicza 24, 85-092 Bydgoszcz, Poland; w.zielinska@cm.umk.pl (W.Z.); mizdebska@cm.umk.pl (M.I.); agrzanka@cm.umk.pl (A.G.)

**Keywords:** piperlongumine, sanguinarine, combination therapy, lung cancer, cell death

## Abstract

Background: Cancers are one of the leading causes of deaths nowadays. The development of new treatment schemes for oncological diseases is an interesting direction in experimental medicine. Therefore, the evaluation of the influence of two alkaloids—piperlongumine (PL), sanguinarine (SAN) and their combination—on the basic life processes of the A549 cell line was considered reasonable. Methods: The aim was achieved by analyzing the cytotoxic effects of PL and SAN and their combination in the ratio of 4:1 on the induction of cell death, changes in the distribution of cell cycle phases, reorganization of cytoskeleton and metastatic potential of A549 cells. The versatility of the applied concentration ratio was evaluated in terms of other cancer cell lines: MCF-7, H1299 and HepG2. Results: The results obtained from the MTT assay indicated that the interaction between the alkaloids depends on the concentration and type of cells. Additionally, the compounds and their combination did not exhibit a cytotoxic effect against normal cells. The combined effects of PL and SAN increased apoptosis and favored metastasis inhibition. Conclusion: Selected alkaloids exhibit a cytotoxic effect on A549 cells. In turn, treatment with the combination of PL and SAN in a 4:1 ratio indicates a synergistic effect and is associated with an increase in the level of reactive oxygen species (ROS).

## 1. Introduction

Currently, in addition to cardiovascular and cerebrovascular diseases, cancers are one of the leading causes of death. Furthermore, lung cancers are at the top of the list of the most frequently diagnosed cancers in the world. The high mortality in patients with lung cancers is associated with several aspects. One of them is the appropriate diagnosis, which is especially difficult in an early stage of the disease. In turn, for patients with a late stage of cancer, metastasis are very often diagnosed. From a clinical point of view, the applied treatment procedure primarily depends on the type of lung cancer. In this context, two main groups may be distinguished: small-cell lung cancer (SCLC) and non-small cell lung cancer (NSCLC). NSCLC accounts for up to 84% of lung cancers, and among them, the division into three subgroups is known: squamous cell carcinoma, large cell carcinoma and adenocarcinoma [1,2,3]. Despite the treatment method used, numerous oncology tests and the extension of modern diagnostic methods, the chosen therapy does not always bring satisfactory results. In this aspect, the specific response and resistance of cancer cells to the treatment methods, including cytostatics (multidrug resistance), is a huge barrier [4,5,6,7,8,9]. On the other hand, the cytotoxic effect of drugs on normal cells and side effects of therapy, as well as the metastatic potential of cancer cells, are also serious limitations of treatments. Based on the mentioned aspects, compounds of natural origin have been a part of the canon of new, more effective treatment alternatives. This trend includes alkaloids, which have become an interesting topic of research due to their easy acquisition and a wide range of actions, as well as very selective effects on cancer cells with minimal cytotoxicity against normal ones [10,11,12].

Alkaloids are a very wide group of compounds, including about 100 thathave been used in natural herbal medicine, as well as conventional treatment. These substances are found in many plants and consumed products, e.g., herbs and spices. In addition to affecting the taste of food, alkaloids can also be important in preventive healthcare. Literature reports indicate the anticancer prosperities of these natural compounds concerning different types of cancers, both in vitro and in vivo. Some of them, e.g., camptothecin—a topoisomerase I inhibitor and vinblastine, a factor inhibiting the formation of a karyokinetic spindle—have already been successfully used as chemotherapeutics [13,14]. Furthermore, alkaloids may sensitize drug-resistant cancer cells to cytostatics and induce cell cycle arrest in various phases, thereby promoting the inhibition of tumor proliferation and metastasis [10]. Numerous studies indicate a wide spectrum of activity of alkaloids, including berberine, oxymatrine, piperlongumine and sanguinarine. The last two substances have strong anticancer properties and, thus, have become a very attractive research target. On the other hand, one of the strategies to improve anticancer treatment regimens may be to combine natural compounds, such as alkaloids characterized by a wide spectrum of actions. The validity of these assumptions is based on the literature reports on their selective actions and minimal impacts on normal cells [10,11,14].

The first of the mentioned alkaloids, piperlongumine (PL/PPLGM), may be isolated from the fruits of long pepper plants (*Piper longum* L.) (Figure 1A). As reported by numerous studies, PL exhibits many anticancer prosperities on several types of tumors, including breast, colon, pancreatic, stomach and prostate cancers, through DNA damage, cell cycle arrest, the inhibition of proliferation and induction of reactive oxygen species (ROS) and cell death [15,16,17,18,19,20,21,22]. In turn, sanguinarine (SAN), originating from *Sanguinaria canadensis* L., presents a wide spectrum of action (antibacterial, antifungal, anti-inflammatory and antiplatelet), including anticancer (Figure 1B) [10,23,24,25]. SAN induces alterations in cell cycle phases and apoptosis in several types of cancer, including breast, prostate, leukemia, pancreatic, melanoma and lung cancer [26,27,28,29,30,31]. The developed combination of PL and SAN is innovative. However, similar mechanisms of action—the induction of apoptosis (but in different pathways) and ROS—were considered as a potential advantage.

The presented study aimed to determine the influence of two alkaloids—PL, SAN and their combination—in a ratio 4:1 on the basic life processes of NSCLC A549 cells. Furthermore, in this study, we report the first experimental evidence on the synergistic action of selected alkaloids on lung cancer.

## 2. Results

### 2.1. The Cytotoxic Effect of Piperlongumine (PL) and Sanguinarine (SAN) Individually and in Combined Treatment on Cell Viability

The 3-(4,5-Dimethylthiazol-2-yl)-2,5-Diphenyltetrazolium Bromide (MTT) assay was used to determine the dose-dependent interactions of PL and SAN alone and their combination on the viability of MRC-5, A549, H1299, MCF-7 and HepG2 cells after 24-h exposure. Additionally, the effect was assessed after 48 and 72 h and the results were presented in the Appendix A. The analysis of the type of drug interactions was carried out according to the Chou-Talalay method [32]. Analysis of MTT results showed that PL and SAN did not induce significant changes in the survival of normal MRC-5 lung cells (Figure 2A,B). Moreover, treatment with the combination of PL and SAN in ratio 4:1 also did not show a cytotoxic effect on normal lung cells (Figure 2C).

As was shown in Figure 2D,E, A549 cells treated for 24 h with PL in the concentration range from 1 µM to 8 µM and SAN at doses from 0.25 µM to 2 µM showed a doses-dependent survival decreases in comparison to control cells. Following treatment of A549 with PL, 99.31% ± 5.15%, 96.6% ± 4.68%, 82.99% ± 8.33%, 63.69% ± 1.57% and 50.38% ± 7.46% of live cells for the concentration range 1-8 µM were observed, respectively (Figure 2D). The SAN doses generated 99.40% ± 5.03%, 98.19% ± 5.12%, 94.89% ± 9.54%, 76.53% ± 3.39% and 30.02% ± 8.39% of live cells for 0.25, 0.5, 1, 1.5 and 2 μM (Figure 2E). In turn, in the case of combined treatment, the percentages of live cells of 81.19% ± 5.79% for 1-μM PL/0.25-μM SAN, 59.10% ± 6.34% for 2-μM PL/0.5-μM SAN, 46.72% ± 9.95% for 4-μM PL/1-μM SAN, 16.9% ± 7.18% for 6-μM PL/1.5-μM SAN and 11.29% ± 3.49 for 8-μM PL/2-μM SAN were noted (Figure 2F). The half-maximal inhibitory concentrations (IC_50_) values for 24-h incubationswere calculated by the CompuSyn program and reached 8.03µM for PL and 2.19 µM for SAN. Analysis of the type of interaction showed a CI value of < 1, which is characteristic for synergism (Figure 2G).

The versatility of the applied concentration ratio was evaluated in terms of other cancer cell lines. The obtained data showed that the H1299 treatment with both selected alkaloids induced a statistically significant and dose-dependent decrease in cell survival, except for the lowest concentration (Figure 3A,B). In turn, the exposure of H1299 cells to a combination of selected alkaloids in ratio 4:1 induced an increase in cytotoxicity, as well as a statistically significant reduction in the percentage of live cells for all combinations in comparison to untreated cells (Figure 3C). However, the analysis of alkaloids’ interactions showed that the treatment of H1299 cells with lower concentrations in the combination indicates an additive effect and, for higher doses, an antagonism (Figure 3D).

In the case of MCF-7 cells, the data presented in Figure 3 indicates a statistically significant dose-dependent decrease in the survival of cells after 24-h exposure for both individual and combined alkaloids. The results obtained for the combination did not differ significantly from those observed following PL or SAN treatments (Figure 3E–G). The analysis of the combination by CompuSyn showed that, for 4-µM PL/1-µM SAN, as well as for other concentrations, the CI value reached >1, which indicated antagonism (Figure 3H).

In turn, the exposure of HepG2 cells to the combination of alkaloids in ratio 4:1 strongly intensified the cytotoxic effect of the compounds, as it induced a greater reduction in the percentage of live cells compared to the individual actions of PL or SAN (Figure 3I–K). The calculated index combination for the combined action of selected alkaloids was CI < 1, which is characteristic for synergism (Figure 3L).

### 2.2. Effect of PL, SAN and the Combination of Alkaloids on Cell Death in the A549 Cell Line

The evaluation of cell death presented that 24-h treatment with selected doses of alkaloids—1-µM SAN and 4-µM PL resulted in a decrease in the percentage of A549 live cells (AV−/PI−) in comparison to untreated cells. For the control (CTRL), 92.09% ± 2.3%, 84.54% ± 3.4% for 1-µM SAN and 73.85% ± 2.4% for 4-µM PL of live cells were noted, respectively. In the case of 24-h incubation with a combination of PL and SAN in ratio 4:1, the decrease in cell viability of the tested A549 cells to 60.59% ± 6.3% was visible (Figure 4A). Figure 4B presents an increase in the percentage of early apoptotic cells (AV+/PI−). The values reached 2.75% ± 0.5% for untreated A549 cells, 4.56% ± 1.0% for 1-μM SAN, 6.19% ± 0.3% for 4-μM PL and 7.92% ± 1.5% for the 4-μM PL/1-μM SAN (Figure 4B). Similarly, in the case of cells with a double-positive signal for annexin V and propidium iodide (late apoptosis; AV+/PI+) treatment with alkaloids and their combination induced an increase in the population from 2.94% ± 1.06% (CTRL) to 5.36% ± 2.78% (1-μM SAN), 11.26% ± 2.33% (4-μM PL) and 21.06% ± 5.36% (4-μM PL/1-μM SAN) (Figure 4C). The analysis of cell death also showed the necrotic population (AV−/PI+) at the range 3.71% ± 1.6% for control A549 cells and 6.44% ± 1.03%, 8.8% ± 0.77% and 11.65% ± 2.98% for 1-μM SAN, 4-μM PL and a combination of the alkaloids in ratio 4:1 (Figure 4D). The reported value for both PL alone and in combination with SAN were statistically significant comparing to the untreated cells.

As mentioned above, the double-staining of annexin V and propidium iodide presented that a 24-h treatment with selected alkaloids and their combination resulted in apoptosis in A549 cells. To further identify the type of induced cell death, the analysis of the level of proteins involved in a particular type of apoptosis pathway was performed. The cytometric measurement of mean fluorescence for Apaf-1 showed a statistically significant increase for A549 cells treated with alkaloids—1.76 ± 0.83-fold for 1-μM SAN and 3.20 ± 1.12-fold for 4-μM PL in comparison to control cells (Figure 4E). Similarly, in the analysis carried out for caspase-12, there was also a statistically significant increase in the average fluorescence intensity of the protein—2.97 ± 1.01-fold and 1.64 ± 0.69-fold in comparison to untreated A549 cells, respectively, for applied doses of SAN and PL (Figure 4F). In both cases, the treatment of A549 cells with 4-μM PL/1-μM SAN did not induce significant changes for individual proteins (increase for Apaf-1 1.07 ± 0.49-fold and, for caspase-12, 1.31 ± 0.75-fold) (Figure 4E,F). Figure 4G shows the results for caspase-8, which indicated that the combined effect of PL and SAN in ratio 4:1 resulted in a 2.86 ± 1.03-fold increase in the average level of fluorescence intensity comparing to the CTRL. In turn, the treatment with individual doses of alkaloids induced a 1.24 ± 0.34-fold and 1.55 ± 0.46-fold increase, respectively, for SAN and PL (Figure 4G). These results allow us to assume that the selected concentration of 4-μMPL induced the intrinsic pathway of apoptosis and 1-μMSAN the stress-induced pathway, while the combination of alkaloids resulted in the extrinsic pathway.

### 2.3. Influence of PL, SAN, and TheirCombination on aCell Cycle Phases Distribution

To assess the effects of PL, SAN and their combination on the basic cellular processes of A549 cells, a cell cycle phases distribution analysis was performed. Treatment with 4-μM PL and a combination with 1-μM SAN induced a statistically significant increase in the percentage of cell populations with DNA content <2N (SubG1) from 0.75% ± 0.29% (CTRL) to 2.39% ± 1.18% and 5.11% ± 1.96% for 4-μM PL and 4-μM PL/1-μM SAN, respectively (Figure 5A). The analysis of the G0/G1 phase showed that 24-h incubation with 1-μM SAN resulted in an increase in the percentage of cells in this fraction from 48.55% ± 4.75% (CTRL) to 59.16% ± 9.75%. Moreover, in other cases, a statistically significant decrease in G0/G1 to 32.65% ± 2.93% (4-μM PL) and 35.15% ± 3.48% (4-μM PL/1-μM SAN) was noted (Figure 5B). Figure 5C shows the percentage of the cell population with DNA content corresponding to the S phase. Treatment with a combination of PL and SAN in ratio 4:1 induced a statistically significant increase in the number of cells in this population compared to the CTRL (from 10.73% ± 3.63% to 14.50% ± 2.99%). In the case of incubation of A549 cells with individual alkaloids, a similar level of the cells in the S phase (10.95% ± 2.97% for 1-μM SAN) and a slight decrease (6.73% ± 2.45% for 4-μM PL) were observed in comparison to the CTRL (Figure 5C). In turn, a 24-h treatment with selected doses of alkaloids caused an increase from 37.26% ± 3.61% (CTRL) to 53.64% ± 1.47% for 4-μM PL and a decrease to 24.89% ± 7.29% for 1-μM SAN in the case of the G2/M phase (Figure 5D). Furthermore, the evaluation of A549 cells with DNA >4N showed that the highest level (7.12% ± 2.36%, statistically significant) was induced by the combination of the alkaloids. While the values of 2.11% ± 0.67% for the control, 3.80% ± 1.07% for 1-μM SAN and 5.79% ± 1.46 for 4-μM PL were noted (Figure 5E). The described distribution of the individual phases of the cell cycle corresponded to the previously observed alterations associated with cell death and changes in the morphology.

### 2.4. Impact of PL and SAN on the Alterations in Morphology and Ultrastructure of A549 Cells

The tests of the quantitative measurement of the percentage of apoptotic cells have been extended to the analysis of alterations induced by PL, SAN and a combination of the alkaloids in ratio 4:1 in the cell morphology and ultrastructure. The evaluation was based on the observation of the morphology with an inverted contrast-phase microscope and a cell nuclei analysis with a fluorescence microscope and transmission electron microscope (Figure 6).

The treatment of cells with 1-µM SAN did not induce any significant changes in the morphology of the A549 cell line. The cells similar to the appearance of the control predominated. Furthermore, a small population with features typical for mitotic catastrophe (multinucleated giant cells) was noticed. The number of cells was not significantly reduced compared to untreated cells (Figure 6B,F). In turn, the incubation with 4-µM PL increased the number of cells with shrunken cytoplasms and cells with outlined features of cell death (cell nucleus fragmentation, apoptotic bodies or chromatin condensation). Moreover, the number of cells observed in the field of view was also reduced (Figure 6C,G). However, the greatest changes in the morphology were noted after using a combination of alkaloids (4-μM PL/1-μM SAN). Furthermore, an increase in the number of shrunken cells with apoptotic bodies was observed. Moreover, cells with one large nucleus or numerous micronuclei were visible (Figure 6D,H).

The analysis of the ultrastructure showed that untreated cells were characterized by homogeneous sizes and regular cell nuclei (Figure 6I). Similar to the effect of SAN on the morphology of A549, the ultrastructure of the cells was not significantly changed after 24-h incubation with this alkaloid (Figure 6J,J′). In turn, following the treatment with 4-µM PL, the increase in the number of altered mitochondria in enlarged cells, as well as the level of cells with visible chromatin condensation in the nucleus and shrunken cytoplasm, were noted (Figure 6K,K′). The combination of two alkaloids resulted in a large population of A549 cells with typical apoptotic features—reduced cell size and shrunken cell nucleus with chromatin marginalization (Figure 6L,L′).

### 2.5. Relationship between the Synergistic Effects of the Alkaloids and ROSFormation

Literature reports indicate that the actions of PL and SAN are associated with ROS induction in various types of cells [17,28]. Thus, in this research on the synergistic action of the compounds, the induction of ROS was also determined. The obtained results indicated a statistically significant increase in the level of average ROS fluorescence intensity at all applied doses in comparison to the CTRL. However, the highest value was recorded for the combined effect of PL and SAN and reached a 7.05-fold increase in the level of ROS compared to the group of untreated cells (15.04% ± 3.62% for 4-µM PL/1-µM SAN). When A549 cells were treated with individual alkaloids, values of 7.87% ± 1.02% and 5.31% ± 1.79% were noted, i.e., 3.7-fold and 2.49-fold increases in ROS levels for 1-µM SAN and 4-µM PL, respectively. During the experiment, as a consequence of using a2-h preincubation with an inhibitor of ROS (*N*-acetyl-l-cysteine; NAC), the partial inhibition of ROS generation was noticed (Figure 7A). To further evaluate whether the cytotoxic effects of tested alkaloids and the synergistic effect of their combination is associated with the production of ROS, after a 2-h pre-incubation with NAC, the double-staining with annexin V and propidium iodide was performed. The results presented that the percentagesof both early and late-apoptotic, as well as necrotic cells,were reduced (data not shown). For the A549 cells treated with the chosen alkaloids with the addition of NAC, a decrease in the population of cells with a positive signal for annexin V (hallmark of apoptosis) to 3.82% ± 0.38% (from 9.64% ± 2.41%) for 1-μM SAN and 12.04% ± 2.25% (from 17.45% ± 2.14%) for 4-μM PL were observed. The greatest reduction was noted for the combination of the selected alkaloids in the ratio 4:1, and it reached 15.25% (from 28.50% ± 7.19% to 13.25% ± 2.09%) (Figure 7B). In addition, to confirm whether the inhibition of the ROS generation reduces the observed effect of PL, SAN and their combinations, a cell cycle analysis was performed. Data presented in Figure 7C indicates that the suppression of ROS formation in cells following 4-μM PL and 4-μM PL/1-μM SAN treatments caused alterations in the number of cells in populations with DNA corresponding to almost all phases of the cell cycle in comparison to A549 cells treated with the selected alkaloids but without the preincubation with NAC. Thus, in the case of the analysis of the S phase, the highest increase in the cell number observed ranged from 7.02% ± 2.43% (A549) to 20.39% ± 0.68% (A549 + NAC) for PLin the dose of 4 μM. Similarly, in the cell fraction for DNA>4N, an increase in the percentage for the treatment with PL from 6.32% ± 1.59% (A549) to 12.24% ± 0.62% (A549 + NAC) and from 7.86% ± 2.98% (A549) to 13.95% ± 0.62% (A549 + NAC) for 4-μM PL/1-μM SAN were noted. In turn, a decrease in the G2/M phase from 53.43% ± 1.37% to 30.71% ± 1.24% and from 41.37% ± 8.16% to 33.39% ± 2.93% for A549 cells without and after the addition of 5-mM NAC and next treated with 4-μM PL were noticed (Figure 7C). The above-mentioned results have suggested that the synergistic mechanism of action of the combination of PL and SAN in ratio 4:1 is associated with ROS generation.

### 2.6. The Effects of PL and the Combination with SAN on Cytoskeletal Proteins in A549 Cells

NSCLC A549 control cells were characterized by a regular and well-developed network of microtubules with visible centrosomes in the perinuclear area (Figure 8A,A′). Following a 24-h treatment with 1-µM SAN, the selected concentration did not induce any significant changes in the β-tubulin organization. The exception was single cells with a phenotype of mitotic catastrophe, where the accumulation of a signal in the form of a ring located around the cell nucleus was observed (Figure 8B,B′). In turn, the treatment with 4-µM PL resulted in the presence of cells with a dispersed microtubule network and point accumulation of β-tubulin in the cytoplasm (Figure 8C,C′). Furthermore, the incubation with a combination of alkaloids in ratio 4:1 caused an increase in the number of shrunken cells with a strong distribution of microtubule fibers and a decrease in the level of visible fluorescence. Moreover, cells with an apoptotic phenotype had a strong β-tubulin point signal (Figure 8D,D′).

The assessment of the actin network indicated that control cells were characterized by an extensive actin cytoskeleton (Figure 8E,E′). As a result of a 24-h incubation with 1-µM SAN, no significant changes in the organization of the microfilament network were observed. A small population of giant cells with stress fibers was noted (Figure 8F,F′). After exposure to 4-µM PL, the accumulation of visible F-actin signals in the form of points located near the cell nucleus was observed. In the case of apoptotic cells, PL induced a strong reorganization, visible as the dispersion of theactin fibers. In addition, cells with stress fibers and actin in the form of foci were also noticed (Figure 8G,G′). In turn, the combination of the two alkaloids intensified the disorders in the organization of microfilaments. Besides the point accumulation of F-actin, a reduction in the visible fluorescence intensity in the cells was observed. Furthermore, numerous stress fibers and cells with a diffused fluorescent signal of labeled protein were observed (Figure 8H,H′).

The last assessed cytoskeleton protein was vimentin. Similarly to the above-described F-actin and β-tubulin, control cells were characterized by a well-developed vimentin network (Figure 8I,I′). Exposure to 1-µM SAN also, in this case, did not induce significant changes, except to shrunken cells, where a decondensed network of vimentin fibers was noted. There were also a few cells with lower levels of florescence intensity. Moreover, cells with the phenotype of mitotic catastrophe and a dispersion of the vimentin network were observed (Figure 8J,J′). In turn, the treatment of A549 cells with 4-µM PL contributed to a gradual lowering of the vimentin fluorescent signal. In this case, cells with visible single fibers radiating throughout the entire cytoplasm and with point accumulation near the nucleus were also noticed (Figure 8K,K′). The combination of PL and SAN induced the reorganization of the vimentin network and an increase in the population of cells with apoptotic features. These cells were smaller in size, and consequently, a reduction in the density of the vimentin, as well as its location in the form of clusters, was visible (Figure 8L,L′).

### 2.7. The influence of PL and Its Combination with SAN Onmigration and the Invasivepotential of A549 Cells

To determine the effects of selected concentrations of PL, SAN and their combination on the metastatic potential of A549 cells, a wound-healing assay was performed. The obtained results showed that the mechanically scratched “wound” in the control cells was completely overgrown after about 29 h. At the same time, the gap was covered 92.46% ± 1.83% and 83.76% ± 2.96% after treatments with 1-µM SAN and 4-µM PL, respectively. The incubation of cells with a combination of alkaloids reduced the metastatic potential compared to the control even further. After 29 h, the scratch was covered 75.66% ± 4.95% (Figure 9A). The presented results indicate that PL at a concentration of 4 µM inhibits the migration of A549 cells to a greater extent than 1-µM SAN. However, the greatest reduction in the cells’ motility was induced by the treatment with a combination of the alkaloids in ratio 4:1 (Figure 9A). In addition, the wound area was also analyzed after 24 h. In untreated cells, the mechanical “wound” was covered 86.40% ± 3.44% while the values of 79.51% ± 2.65% and 59.01% ± 2.56% were noted for the doses of SAN and PL, respectively. The treatment of A549 cells with 4-µM PL/1-µM SAN induced a wound area reduction of 51.84% ± 2.45% (Figure 9B). The values obtained for PL and the combination with SAN showed statistical significance in comparison to the control.

The effects of PL and SAN individually and in combination on the metastatic potential of A549 cells was also tested using migration and invasion assays. The assessment of the migration potential showed that, following a 24-h treatment with selected concentrations of PL, SAN and their combination, the number of cells observed on the outside of the insert decreased. In the control, 935.7 ± 120.5 cells were visible. After treatment, the numbers of cells with high migratory potential were reduced to 843 ± 46.04 for 1-µM SAN and to 324.0 ± 57.94 for 4-µM PL. The smallest number of cells in the field of view was confirmed after applying a combination of the alkaloids (234.5 ± 64.41) (Figure 9C).

Similar results were obtained in the invasion test. The number of cells observed on the outside of the insert in the tested samples was lower than that obtained in the migration test. However, also, in this case, it decreased in comparison to the control. For the control, 666 ± 133.8 cells were observed, while the number decreased to 602.7 ± 151.1 and 248.3 ± 99.07 for 1-µM SAN and 4-µM PL, respectively. When the alkaloids were combined in the ratio 4:1, 159 ± 33.25 cells were noticed (Figure 9D).

In order to confirm the results described above, the analysis was enriched with the assessment of the level of markers of the epithelial-to-mesenchymal transition (EMT), the process related to the metastatic potential of cells. The evaluation was based on data obtained from the cytometric measurements of the average fluorescent intensity of the main determinants of this process—N- and E-cadherin and vimentin. The evaluation of N-cadherin indicated a decrease in the level of the fluorescence intensity of the protein in A549 cells. After treatment with 1-μM SAN, the observed decrease was not significant (0.97 ± 0.23-fold). In turn, the effect of 4-μM PL and the combination of the alkaloids in ratio 4:1 resulted in a statistically significant reduction compared to untreated cells, as 0.83 ± 0.07-fold and 0.72 ± 0.18-fold reductions were noted, respectively (Figure 9E).

In the case of the cytometric measurement of E-cadherin, the analysis showed an increase in the average level of the fluorescence intensity of the protein after a 24-h incubation with PL and SAN individually and in combination compared to the CTRL. The increase ranged from 1.17 ± 0.12-fold to 1.27 ± 0.14-fold and 1.49 ± 0.21-fold for 1-μM SAN, 4-μM PL and 4-μM PL/1-μM SAN, respectively (Figure 9F).

The last of the analyzed markers was vimentin. The earlier analysis of fluorescence staining showed a reorganization of the protein network and a decrease in the visible fluorescence intensity. These observations were confirmed by the cytometric evaluation of the average level of the fluorescence intensity. The treatment with the selected alkaloids induced a statistically significant decrease in the level of vimentin in comparison to the untreated cells (0.83 ± 0.07-fold for 1-μM SAN and 0.64 ± 0.14-fold for 4-μM PL). Moreover, the combination of PL and SAN caused a significant (0.56 ± 0.14-fold) reduction of the mean fluorescence level of the evaluated protein (Figure 9G).

Due to the importance of actin filaments in the cell migration [33], the analysis was also enriched with cytometric measurements of the protein level. Following a 24-h treatment of NSCLC cells with individual alkaloids induced a decrease in theaverage level of F-actin fluorescence intensity, which was 0.98 ± 0.13-fold and 0.74 ± 0.1-fold for SAN and PL, respectively. In the case of incubation with the alkaloids in the combination, a statistically significant decrease in F-actin compared to the CTRL was observed (0.68 ± 0.14-fold) (Figure 9H).

## 3. Discussion

Civilization diseases, including cancers, are currently one of the biggest problems of the aging society. It is estimated that, over the next decades, cancer mortality will dramatically increase [34]. Unfortunately, despite the development of diagnostic methods and modern lung cancer treatments, the results are still not fully satisfying [34,35,36]. A poor prognosis is associated with the resistance of this type of tumor to the therapy and undesirable effects caused by cytostatics. In addition, the high invasiveness of lung cancer cells contributes to the formation of new foci of cancer (metastasis). The above-mentioned factors constitute a barrier for effective oncological therapy. Therefore, new possibilities and drugs that are highly selective in their actions on lung cancer cells and nontoxic for normal cells are the important goals of experimental medicine. Besides current trends in basic science and literature reports, in this study, we determined the effects of two natural alkaloids—SAN and PL, as well as their combination—on the basic life processes of A549 NSCLC cells.

The first step of our investigation was an evaluation of the cytotoxicity and the type of interaction between PL and SAN in the A549 cell line and other types of cancer cells—MCF-7, HepG2, H1299 and normal lung cells MRC-5. Results obtained from the MTT assay allowed to determine the IC_50_ for the A549 cell line, which reached the values of 8.03 µM for PL and 2.19 µM for SAN, respectively. Various doses and values of the IC_50_ of the selected alkaloids have been reported in the literature. In the studies provided by Zheng et al. and Zhang et al. on two types of lung cancer cell lines (A549 and NCI-H460), it was shown that the IC_50_ for PL ranged from 13 to 15 µM [36,37]. Similar results were obtained by Wang et al., who confirmed a high cytotoxicity and dose-dependent reduction of A549 survival following a treatment with PL [38]. The effects of the alkaloidhas been evaluated on cancers cells, including breast (MCF-7); colon (HT29, HCT116 and SW620); prostate (PC-3, DU-145 and LNCaP) and high-grade glioma, where the IC_50_ ranged between 5 to 20 µM concentrations [18,19,20,21,22]. On the other hand, several studies showed the nontoxic effects of PL on normal cells [39]. Furthermore, no reduction in the survival of normal cells following the SAN treatment was confirmed [40]. Similar results were obtained in the present study, where the treatments with selected concentrations of PL and SAN and their combination were applied in the case of the lung fibroblast MRC-5 cell line, as no significant changes in the survival of the cells were noted. These observations confirmed the selective action of PLand SAN. Recent scientific reports also indicated a highly cytotoxic effect of SAN on various type of cancers, which conformed to the achieved IC_50_ at low concentrations of the described compounds [27,28,29,41]. In addition, dose- and time-dependent reductions in the cell viability after the treatment with SAN have been reported for cell lines such as HT-29 (colon cancer), K1735-M2 (melanoma), HCT-116 (colorectal), A549 (lung cancer) and hematopoietic cancer cell lines [28,29,30,31,40,42]. According to the literature, the IC_50_ for SAN-treated NSCLC cells ranged from 2 to 15 μM. The above-mentioned concentration of both PL and SAN, which induced a 50% reduction in the survival of lung cancer cells, showed higher values than the doses used in the present study [43]. The observed differences may result from several aspects. Attention should be paid to the source of the compound and cell confluence, which determines their individual response to various factors, including cytotoxic substances. In the literature, this phenomenon is widely known and referred to as the confluence-dependent resistance (CDR) [44,45]. The MTT test results presented in the paper also enabled to distinguish the interaction of the studied alkaloids with the median effect method developed by Chou and Talalay [46]. It is the most commonly used algorithm for assessing the interactions between compounds, which is also used for pharmacological modeling in chemotherapy [47]. In this study, cells of the A549 line were treated with PL and SAN in the ratio 4:1, as the interaction analysis indicated their synergism. However, the combination of the alkaloids showed a slight cytotoxic effect on normal lung cells (MRC-5). The versatility of the combined PL and SAN actions was also tested for other types of cancers. The analysis showed that the effects of the alkaloids and their interaction depend on the type of tumor. This is confirmed by the fact that the use of a combination of the same concentrations of PL and SAN on different cancer lines showed both strong synergism (HepG2) and antagonism (MCF-7 and H1299). The observed effect can be explained by the individual response of cancer cells to selected alkaloids. This fact is connected with the occurrence of numerous biochemical differences between them and their varied genetic conditions, which affect the activation of different cellular response pathways to the same factor [48]. This is the first report regarding the combined actions of selected alkaloids on cancer cell lines. In turn, the results obtained for PL and SAN individually are confirmed by literature reports that present a wide spectrum of activities of the above-mentioned natural compounds [10,41,49]. Besides, the clinical potential of the alkaloids may be demonstrated by the possibility of PL and SAN applications in the forms of combined therapies with conventional cytostatics. Moreover, the compounds are characterized by the high efficiency of the actions, both in vitro and in vivo [22,50,51]. Roh et al. indicated that the simultaneous use of PL and cisplatin in head and neck cancer cells (HNC) effectively increased the drug’s cytotoxicity, while the use of this synergistic combination in an animal model reduced the tumor size compared to the use of the drugs individually [52]. Similar observations were made by Piska et al., who treated prostate cancer cells (DU-145) with a combination of PL and doxorubicin (DOX) [53]. In turn, Chen et al. also presented that treatment of triple-negative breast cancer cell lines (MDA-MB-231 and MDA-MB-453) with PL and DOX resulted in synergism [54]. Additionally, literature reports also indicated the possibility of combining PL with other substances of natural origins. The available data indicate that treatments with pancratystatin (PST) and PL cause areduction in the survival of pancreatic cancer cells [55]. Similar combinations with cytostatics have been studied in the case of the second alkaloid used—SAN. Eid et al. showed that a combination of SAN and DOX enhanced the effects of the drugs on colon adenocarcinoma cells (Caco-2) [56]. It is also worth mentioning that doses of the alkaloids (4-μM PL and 1-μM SAN) applied in this study were selected based on available literature data where the compounds were used in in vitro and in vivo studies.

Data obtained during the analysis of apoptosis indicate that the mechanism of action of PL, SAN and their combination is associated with the induction of a programmed type of cell death. Identification of the apoptosis pathway showed that the treatment with 1-µM SAN promoted stress-induced apoptosis (an increase of the casapse-12 level), while 4-µM PL activated the mitochondrial pathway (higher level of Apaf-1). However, after a 2-h incubation of A549 cells with a combination of the selected alkaloids in ratio 4:1, an increase in the level of caspase-8, necessary for the proper course of the receptor pathway, was noted. The results described above correspond with reports of the world literature, where various types of cancer cells’ deaths activated by alkaloids have beenpresented [18,27,28,29,57]. Wang et al. suggested that the treatment of A549 cell lines with PL at concentrations of 6 µM and 10 µM promotes apoptosis, with the decrease in Bcl-2 levels in favor of Bax. Furthermore, the mentioned group of scientists, by measuring proteins characteristic for autophagy (LC3-III), showed that PL can also induce this type of death [38]. Similarly, studies by Xiong et al. have shown that PL at ranges from 10 to 20 µM in primary bone marrow-derived mononuclear cells (BMMNCs) also induce dose-dependent apoptosis or autophagy [58]. Similar observations were made by Makhovet al., who also identified autophagy in kidney (HEK-293T), breast (MCF-7) and prostate (PC-3) cancer cell lines incubated with 10-µM PL [59]. The obtained results enrich the above literature reports, with the possibility of PL activating another type of death in A549 NSCLC cells—mitotic catastrophe. This type of death was defined in 2010 and is characterized by disturbances in the G2/M phase checkpoints and an increase in the percentage of cells characterized by polyploidy (<4N DNA), as well as the formation of large cells with one giant nucleus (MONGC, mononucleated giant cells) or several micronuclei (MNOC, multinucleated giant cells) [60]. The analysis of changes in the phases of the cell cycle, as well as the morphology, applied in this study, with particular emphasis on cell nuclei, indicates that 4-µM PL induces also an increase in the number of cells with a mitotic catastrophe phenotype. In the case of cell death caused by SAN, most literature reports indicate apoptosis as the main death mechanism promoted by the alkaloid, which correlates with our results. Gu et al. (2015) observed SAN-induced changes in the level of the ROS and the involvement of the endoplasmic reticulum in promoting cell death [61]. The data obtained suggest that 1-µM SAN may promote stress-induced apoptosis, the activation of which is closely related to the above-mentioned changes. Additionally, research groups suggested that the apoptosis pathway induced by SANin lung cancer cells depends on the dose of the alkaloid [61]. Most often, only two pathways of programmed cell deaths are described—intrinsic and extrinsic. However, the stress-induced pathway may be closely related to the executive phase of the above apoptosis pathways, while the concentration used may be the key in explaining the observed differences [62,63]. The literature indicates that SAN favors apoptosis not only in the A549 cells but, also, in HCT-116 (colorectal cancer) and MG-63 (osteosarcoma) cell lines [28,63]. In addition, SAN-induced changes in the cell cycle distribution noted in the study indicated anarrest in the G0/G1 phase and correlated with observations made by other research groups [40,43]. The developed combination of PL and SAN in ratio 4:1 promoted a greater increase in the percentage of apoptotic cells in comparison to individual doses of alkaloids. These changes were observed during the analysis of apoptosis, cell cycle phases and at the level of a light microscope.

Furthermore, available literature data indicate that the actions of PL and SAN may be associated with the formation of the ROS in various types of cancers cells [17,28,61,62]. In our study, we also decided to assess whether the effect of a combination of the alkaloids is connected with the production of the ROS. Over the years, the importance of the ROS in biological processes has been well-documented. It is known that the accumulation of the ROS in cells contributes to changes in biological structures (proteins, lipids and DNA), which, in turn, lead to disorders of the cell metabolism and its death [64,65]. The use of 2-h preincubation with NAC (ROS inhibitor) showed a reduction in the number of ROS-positive cells. Moreover, the pretreatment of cells with NAC followed by the analysis of the cell death showed a decrease in the percentage of apoptotic cells after the treatment with a combination of PL and SAN in ratio 4:1. The obtained results suggested that the synergistic effect of the selected alkaloids may be associated with the induction of the ROS and, consequently, the promotion of the external pathway of apoptosis.

The results described above correspond to the observations regarding changes in the morphology at the ultrastructural level and reorganization in the main cytoskeleton proteins, such as microtubules, vimentin and F-actin. It is known that cytoskeleton is important in basic cellular processes. Furthermore, this is a particularly interesting aspect of the study, as few reports describing the properties of selected alkaloids in the context of changes in the cytoskeleton are available. Our analysis showed that 1-µM SAN did not induce significant alterations in the organization of the cytoskeletal network, except for single apoptotic cells where condensation of the assessed proteins was visible. Opposite results were noted by Wang et al., who suggested that treatments with SAN in MCF-7, HeLa and U2OS cells result in significant changes in the structure and distribution of the microtubule network in the cells. Moreover, the accumulation of the β-tubulin signal in the area of the cell nucleus was observed [38]. The differences between the observations may result from the discrepancy of the SAN concentration and the type of cancer line used. In the context of cytoskeletal proteins, a greater effect was induced by the incubation of A549 cells with 4-µM PL. Meegan et al., based on the fluorescent stainingof microtubules in MCF-7 cells treated with PL, presented extensive fiber reorganizations [66]. Similar observations were noted by Liu et al., who treated bladder cancer cells (T24 and BIU-87) with PL in the concentration range of 2.5–20 µM [67]. Furthermore, Gagat et al. reported that treatments with 2-µM and 4-µM PL in A549 cells caused changes in the actin cytoskeleton [68]. In turn, a reduction in the level of vimentin was observed in PL-treated oral cancer cells (CGHNC8) [69]. The combination of the alkaloids used in this study caused an enhancement in the reorganization of the vimentin, actin and microtubule networks, which is probably associated with an increase in the cytotoxic effects of the alkaloids, as well as changes at the level of A549 cell proliferation and death.

Both vimentin and F-actin are proteins necessary for the biological process of EMT [70]. The reorganization of these cytoskeleton components observed in the presented work draw attention to the properties of the alkaloids in the context of the inhibition of the metastatic potential of cancer cells. In oncology, an extremely important problem remains metastasis, as this phenomenon often contributes to a high mortality among patients. Considering this, the effects of PL, SAN and their combination on the metastatic potential of A549 NSCLC cells were also assessed. This was achieved using standard methods recommended for the analysis of the EMT, such as a wound-healing assay and migration and invasion tests, as well as measuring the level of markers of this process—vimentin, N- and E-cadherins. It is known that the EMT is conditioned by numerous biochemical and morphological changes of cells, which are effectively used in its identification, as well as determining the properties of potential compounds/drugs in the context of metastasis. The results obtained in this study showed that a 24-h incubation of the A549 cell line with 1-µM SAN and 4-µM PL reduced the migration and invasive potential of the cells. The above-described effect was even intensified after the treatment of NSCLC cells with a developed combination of PL and SAN in ratio 4:1. Park et al., based on the experiments carried out using A549 and MCF-7 cell lines, showed that treatments with PL in the 0.5–10 µM doses range reduced the migration and invasive properties of the selected cancer cells. They also suggested that the inhibition of the EMT by PL may be associated with the effects on the TGF-β/Snail1/Twist pathway, which result in the regulation of the E-cadherin expression level [71]. Data confirming the antimigratory and anti-invasive effects of PL have also been described concerning lung and colorectal cancers [36,72]. The world literature have also reported the use of SAN as an inhibitor of the EMT in cancer cells. Studies carried out by Choi et al. (2008) on breast cancer cells (MDA-MB-231) treated with SAN suggest that the reduction of their metastatic potential is associated with the regulation of the matrix metalloproteinases (MMPs) expression, which is the essential element of the process [73]. In turn, Park et al. showed that SAN can reduce the migration of cancer cells, not only via MMP regulations but, also, by blocking the NF-κβ, Akt and ERK1 signaling pathways [74]. The effects of PL and SAN on the migration and invasion of highly aggressive A549 cells observed in our study correspond to the literature reports cited above. Based on the obtained data, it can be suggested that the antimetastatic potential of alkaloids is related to the reorganization of the actin and vimentin networks, which are important for the motility and EMT. Additionally, the enhanced inhibition of cell movements following the combined actions of PL and SAN in ratio 4:1 was observed. Results confirmed the synergistic effects of selected alkaloids in comparison to individual concentrations of PL and SAN on the non-small lung cancer A549 cell line also in the case of cell motility.

## 4. Materials and Methods

### 4.1. Cell Culture and Treatment

The NSCLC A549 cell line was kindly provided by P. Kopiński, Ph.D.(Department of Gene Therapy, Ludwik Rydygier Collegium Medicum in Bydgoszcz, Faculty of Medicine, Nicolaus Copernicus University in Toruń, Poland) and grown as a monolayer in standard culture conditions (5% CO_2_, 37 °C, 95% air atmosphere). The A549 cells were cultured in tissue culture flasks and 6-or 12-well plates in Dulbecco’s modified Eagle’s medium (DMEM; Lonza Group, Ltd., Basel, Switzerland) supplemented with 10% fetal bovine serum (FBS; Sigma-Aldrich, Merck KGaA, Darmastadt, Germany) and 50-μg/mL gentamycin (Sigma-Aldrich). In the design of the experiment, the MTT assay was carried out using combinations of PL, SAN and other alkaloids (aconitine and oxymatrine; data not shown). However, the most favorable interaction spectrum corresponding to synergism was estimated only for the combination of PL and SAN. The concentrations used in the study were estimated experimentally based on the available literature data and MTT assay.

To assess the cell viability (MTT assay), cells were treated with various concentrations of natural alkaloids dissolved in dimethylsulphoxide (DMSO) (Sigma-Aldrich): PL (1, 2, 4, 6 and 8 µM; Abcam, Cambridge, United Kingdom); SAN (0.25, 0.5, 1, 1.5 and 2 µM; Abcam) and their combination in ratio 4:1 for 24 h. For other experiments, A549 cells were treated with 4-µM PL, 1-µM SAN and 4-µM PL/1-µM SAN. The control cells were cultured in the same conditions but without treatment. In addition, MRC-5, H1299, HepG2 and MCF-7 cell lines were used to assess the cytotoxicity of the chosen alkaloids. All cell lines were also grown as monolayers in standard culture conditions and recommended media (H1299—Roswell Park Memorial Institute (RPMI) 1640 and HepG2—Eagle’s Minimal Essential Medium (EMEM), MCF-7 and MRC-5—Minimal Essential Medium (MEM); Lonza Group, Ltd.). Additionally, all cell lines were tested for *Mycoplasma* based on the rapid staining with 4′,6-diamidino-2-phenylindole (DAPI) (Sigma-Aldrich), and the results were found to be negative [75].

### 4.2. Cell Viability and Type of Drug Interactions

The MTT (thiazolyl blue tetrazolium bromide; Sigma-Aldrich) colorimetric assay was used to analyzethe cytotoxicity of the selected natural alkaloids. In short, cells were grown as a monolayer on 12-wells plates and then treated with PL at concentrations ranging from 1 to 8 µM, SAN at doses from 0.25 to 2 µM and combinations of the natural agents in the ratio 4:1 for 24 h. The MTT analysis was also performed for 48 h and 72 h (Appendix A). Then, cells were washed with phosphate buffered saline (PBS) and incubated with the MTT working solution (powder was dissolved to the concentration 5 mg/mL in PBS and diluted with DMEM without phenol red in the ratio 1:9) for 3 h in standard culture conditions (37 °C, 5% CO_2_, 95% air atmosphere). Next, purple formazan crystals (resulted from the reduction of MTT by living cells) were dissolved in 2 mL of propan-2-ol (10 min, 37 °C; POCH S.A., Gliwice, Poland). Dye absorbance was measured by a spectrophotometer at a wavelength of 570 nm (Spectra Academy, K-MAC, Daejeon Korea). The cell viability was calculated in the reference to the control (estimated as 100%) and presented in the form of graphs. The results obtained from MTT were also used to categorize the type of effect of the PL/SAN combination on A549; MRC-5 and other selected cancer cell lines (H1299, HepG2 and MCF-7). The analysis was based on the combination index (CI) method of Chou-Talalay and recommended CompuSyn software [46,76]. Analysis by the Chou-Talalay method enabled the assessment of the following interactions of the compounds—synergism (CI < 1), additive effect (CI = 1) and antagonism (CI > 1) [32].

### 4.3. Cell Death

The double-staining with annexin V/propidium iodide (AV/PI) was used to determine the type of cell death. The cell populations were evaluated based on the cell membrane permeability and expression of phosphatidylserine and classified as live (AV−/PI−), early-apoptotic (AV+/PI−), late-apoptotic (AV+/PI+) and necrotic (AV−/PI+). The A549 cells were grown as a monolayer on 12-well plates and treated with 4-µM PL or/and 1-µM SAN for 24 h. In the next step, cells were harvested with 0.25% trypsin (37 °C, 5 min; Sigma-Aldrich) and centrifuged for 8 min at 500× *g*. The Tali Apoptosis Assay Kit—annexin V Alexa Fluor 488 and propidium iodide (Invitrogen and Thermo Fisher Scientific, Inc., Waltham, MA, USA) was used according to the manufacturer’s instructions. The control and treated cells were suspended in 100 µL of annexin-binding buffer (ABB) and incubated with 5 µL of annexin V Alexa Fluor 488 for 20 min (room temperature (RT), dark). After centrifugation (5 min × 300× *g*) and suspension in ABB (100 µL), 1-µL PI was added to each sample for 3 min (RT, dark). The 200 µL of stained cells were loaded to 96-wells plate and analyzed using a Guava easyCyte 6HT-2L Benchtop Flow Cytometer (Merck KGaA) and FlowJo vX0.7 software (FlowJo LLC, Ashland, OR, USA).

To evaluate the type of cell death pathway induced by selected alkaloids and their combination in the ratio 4:1, the cytometric analysis of cell death-related proteins (caspase-8, caspase-12 and Apaf-1) was performed. For this purpose, A549 cells were cultured on 6-well plates and incubated with 4-µM PL, 1-µM SAN and 4-µM PL/1-µM SAN for 24 h. After harvesting with 0.25% trypsin (37 °C, 5 min; Sigma-Aldrich), centrifugation (8 min × 500× *g*) and rinsed with PBS, cells were fixed with Cytofix/Cytoperm (20 min, RT; BD Falcon, Franklin Lakes, NJ, USA). Then, cells were suspended in 80% methanol (POCH S.A) and stored at −20 °C untilanalysis. Next day, the control and treated cells were washed with PBS (5 min, RT), and the nonspecific background was blocked with 4% bovine serum albumin (BSA; Sigma-Aldrich) for 20 min. For the detection of selected proteins, the primary rabbit caspase-8, caspase-12 and Apaf-1 antibodies (dilution 1:100 in BSA, 30 min, RT; Thermo Scientific, Rockford, IL, USA) and secondary anti-rabbit AlexaFluor 647 antibody (dilution 1:500 in PBS, 30 min, RT, dark; Invitrogen and Thermo Fisher Scientific, Inc.). After incubation with the antibodies, cells were washed with PBS (3 × 5 min, RT) and centrifuged (5 min × 300× *g*). The measurement was performed using a Guava easyCyte 6HT-2L Benchtop Flow Cytometer (Merck KGaA) and FlowJo vX0.7 software (FlowJo LLC).

### 4.4. Cell Cycle

To detect the alterations in cell cycle phases induced by alkaloids at concentrations 4-µM PL, 1-µM SAN and the combination of the agents in the ratio 4:1, the Tali Cell Cycle Kit (Invitrogen and Thermo Fisher Scientific, Inc.) was used. The A549 cells were cultured on 12-well plates and treated with selected alkaloids (4-µM PL, 1-µM SAN and 4-µM PL/1-µM SAN) for 24 h. Following trypsinization and centrifugation (8 min × 500× *g*, RT), cells were fixed in 1-mL cold 80% ethanol and stored at −20 °C for 24 h. In the next step, control and treated cells were washed with PBS and centrifuged (7 min × 500× *g*, RT). The 200 µL of Tali Cell Cycle solution was added to each sample; then, cells were incubated for 30 min (RT, dark). The 200 µL of stained cells were loaded onto a 96-wells plate and analyzed usinga Guava easyCyte 6HT-2L Benchtop Flow Cytometer (Merck KGaA) and FlowJo vX0.7 software (FlowJo LLC).

### 4.5. Morphology and Ultrastructure of Cells

To see whether PL, SAN and the combination of the alkaloids in the ratio 4:1 affect the alterations in the morphology and ultrastructure of A549 cells, an inverted contrast-phase microscope, fluorescence microscope and transmission electron microscope were used. Cells were seeded on 6-well plates and treated with 4-µM PL and/or 1-µM SAN. Following a 24-h incubation, material was fixed with 4% (*w/v*) paraformaldehyde (PFA, 20 min, RT; Sigma-Aldrich) and washed with PBS (3 × 5 min, RT). The slides were evaluated using a TE100-U inverted contrast-phase microscope (Nikon, Tokyo, Japan) and documented with a DS-5Mc-U1 CCD camera with Nikon Imaging Softwar (NIS) Elements 4.0 software (Nikon). Additionally, to analyze alterations in the cell nuclei, A549 cells were grown on glass cover slides in 12-well plates. After the 24-h treatment and fixation, cells were stained with DAPI (1:20,000 in distilled H_2_O, darkness, RT, 10 min; Sigma-Aldrich) and evaluated by a Nikon Eclipse E800 fluorescence microscope and NISElements program 4.0 (Nikon). In the aim to assess the ultrastructure, control and treated cells were seeded on 6-well plates, harvested (0.25% trypsin, 37 °C, 5 min; Sigma-Aldrich;) and fixed with 3.6% (*v/v*) glutaraldehyde solution (Polysciences, Warrington, PA, USA) prepared in 0.1-M cacodylate buffer (pH 7.4; Roth, Karlsruhe, Germany). Then, after centrifugation (5 min × 300× *g*), cells were washed with 0.1-M cacodylate buffer, and a fibrin clot (Sigma-Aldrich) was prepared from the pellet. In the next step, 1% (*w/v*) osmium tetroxide (Serva, Heidelberg, Germany) prepared in 0.1-M cacodylate buffer was used to postfix the cells (1 h, RT), and the next step was dehydration in increasing concentrations of ethanol (30–90%) and acetone (90–100%) (POCH S. A.). Then, the material was infiltrated with epoxy resin (Epon 812; Roth) with hardeners 2-Dodecenylsuccinic acid anhydride (DBA; Roth) and Methyl nadic anhydride (MNA;Roth) and pure acetone (resin:acetone 1:3; 1:1 and 3:1) and, finally, embedded in gelatin capsules (Ted Pella, Inc., Redding, CA, USA) filled with pure epoxy resin with DMP-30 (Roth). The process of resin polymerization took place over several days: 24 h in 37 °C, 120 h in 65 °C—after which, the prepared capsules were cut on an ultramicrotome Reichert Om U3 (Reichert, Wien, Austria). Ultra-thin sections were placed on copper grids and contrasted with a 1% (*w/v*) uranyl acetate solution (Ted Pella, Inc.). Then, the material was evaluated using a transmission electron microscope JEM-100CX (JEOL, Tokyo, Japan).

### 4.6. Evaluation of the ROS Levels

To understand the cytotoxicity mechanism of PL, SAN and the combination of the compoundsin the ratio 4:1, the alterations in the formation of the ROSwere examined. For this purpose, a CellROX green reagent (Invitrogen and Thermo Fisher Scientific, Inc.) was used according to the manufacturer’s protocols.The A549 cells were grown in 6-well plates and treated with a selected concentration of PL, SAN and the combination of the alkaloids. The cells described as the positive control were incubated with H_2_O_2_ (15 min, 37 °C; Sigma-Aldrich), while the negative control cells were preincubated with 5-mM NAC (2 h, 37 °C; Sigma-Aldrich) and treated with 4-µM PL, 1-µM SAN and 4-µM PL/1-µM SAN for 6h. Following harvesting (0.25% trypsin, 5 min, 37 °C) and washing with PBS, cells were centrifuged (8 min × 500× *g*) and suspended in 1 mL of DMEM without phenol red (Lonza Group, Ltd.) in the addition of 2-µL CellROX green reagent (30 min, 37 °C, dark). In the next step, cells were centrifuged (5 min × 300× *g*) and rinsed with PBS. The 25 µL of stained cells were loaded to TaliCellular slides and analyzed with a Tali Image-Based Cytometer (Invitrogen and Thermo Fisher Scientific, Inc.). The geometric mean of the green fluorescence intensity was analyzed using FlowJo vX0.7 software (FlowJo LLC).

### 4.7. Fluorescence Staining of F-Actin, Vimentin and β-Tubulin

For the detection of alterations in the main cytoskeletal proteins (vimentin, F-actin and β-tubulin), fluorescence staining was used. Firstly, A549 cells were seeded on glass coverslips in 12-well plates. After a 24-h treatment with 4-µM PL, 1-µM SAN and 4-µM PL/1-µM SAN and a fixation with 4% PFA (20 min, RT; Sigma-Aldrich), cells were washed with PBS (3 × 5 min, RT) and permeabilized with 0.25% TritonX-100 (5 min, RT; Serva). After blocking in 1% (*w/v*) BSA (30 min, RT; Sigma-Aldrich), cells were incubated with AlexaFluor 488 conjugated with phalloidin (dilution 1:40 in PBS, 20 min, RT, dark; Invitrogen and Thermo Fisher Scientific, Inc.) for F-actin staining. To label vimentin, the primary mouse anti-vimentin antibody (dilution 1:70 in BSA, 1 h, RT; Sigma-Aldrich) and secondary anti-mouse antibodyAlexa Fluor 488 (dilution 1:100 in PBS, 1 h, RT, dark; Invitrogen and Thermo Fisher Scientific, Inc.) were used. In the case of β-tubulin staining, A549 cells were prefixed with a bifunctional protein crosslinking reagent dithiobis(succinimidyl propionate) (DTSP; 1-mM 3,30-dithiodipropionic acid, diluted 1:50 in Hank’s balanced salt solution, 10 min, RT; Sigma-Aldrich) permeabilized in Tubulin Suspension Buffer (TSB, 0.5% Triton X-100, 10 min, RT; Serva) in a microtubule-stabilizing buffer (MTSB, 1-mM ethylene glycol bis(beta-aminoethyl ether)tetraacetic acid (EGTA), 10-mM piperazine-N,N′-bis(2-ethanesulfonic acid) (PIPES) and 4% poly(ethylene glycol); 10 min; RT; Sigma-Aldrich), with the addition of DTSP. After a series of washing with TSB (3 × 5 min), 4% PFA in MTSB was added to cells for 20 min (RT; Sigma-Aldrich). In the next step, cells were incubated with a mouse monoclonal antibody against β-tubulin (dilution 1:65 in BSA, 1 h, RT; Sigma-Aldrich) and anti-rabbit antibodyAlexa Fluor 594 (dilution 1:200 in PBS, 1 h, RT, dark, Invitrogen and Thermo Fisher Scientific, Inc.).The nonspecifically bounded and unbounded antibodies were removed by washing three times with PBS (5 min, RT). Cell nuclei were stained by standard DAPI-labeling (dilution 1:20,000 in distilled H_2_O, RT, dark; Sigma-Aldrich). After a series of PBS washes (3 × 5 min, RT), the preparations were mounted using Aqua-Poly/Mount (Polysciences) and evaluated by a Nikon Eclipse E800 fluorescence microscope and NISElements 4.0 software (Nikon).

### 4.8. Migration and Invasion

To determine the individual and combined effects of selected alkaloids on the migration and invasion potentials of A549 cells, scratch wound-healing, cell invasion and migration assays were used. For wound healing, cells were cultured on 6-well plates, and then, a monolayer (100% of confluence) was mechanically scratched with sterile plastic tips. The “wound” was rinsed with PBS, and DMEM with a lower concentration of FBS (1%; Lonza Group, Ltd.) was added. Next, cells were treated with PL and/or SAN (referred to as 0h). The observation was performed using a TE100-U inverted contrast-phase microscope and archived using a CCD camera DS-5Mc-U1 and NIS Elements software version 3.30 (Nikon). The data was collected at several time points (3 h, 6 h, 12 h, 24 h, 29 h, 34 h and 36 h) until complete “wound” closure in the case of the control cells. The area of the “wound” in other samples was measured by ImageJ software (Ver1.45s; National Institutes of Health, Bethesda, MD, USA). In the case of the cell migration and invasion assay, the Transwell inserts (24-well polycarbonate filter inserts, 8-μm pore size; Corning Inc., Corning, NY, USA) without and with a layer of 100-µL Matrigel (2 mg/mL in serum-free DMEM; Corning) were used. After a 24-h incubation with PL, SAN and their combination in the ratio of 4:1, cells were seeded at the density of 2.5 × 10^5^ in 350 μL of DMEM without FBS (Lonza Group, Ltd.) to the upper compartment of the insert. DMEM with 15% FBS was added into the lower chambers (750 µL) as a chemoattractant. After 24 h, cells from the upper side of the insert were removed with cotton swabs, and those which migrated on the underside were fixed with 3.6% PFA (20min, RT; Sigma-Aldrich), rinsed with PBS and methanol (20 min, RT; POCH S. A.) and stained with 0.4% crystal violet (dilution in 2% ethanol, 20 min, RT). Analyses of the study in both cases (migration and invasion assays) were based on cell counts from 5 random fields for each sample photographed using a light microscope Nikon Eclipse E800 (Nikon) and CCD camera DS-5Mc-U1 and NIS Elements software version 3.30 (Nikon). The results were presented in the form of graphs and evaluated in reference to the values obtained for the untreated cells. Additionally, to expand the assessment of the potential antimetastatic effects of PL and/or SAN on highly aggressive A549 cells, the cytometric analysis of EMT-related proteins (vimentin and E- and N-cadherin) and F-actin was performed. For the labeling of vimentin and F-actin, A549 cells were cultured on 6-well plates and incubated with 4-µM PL, 1-µM SAN and 4-µM PL/1-µM SAN for 24 h. After harvesting with 0.25% trypsin (37 °C, 5 min; Sigma-Aldrich), centrifuging (8 min × 500× *g*) and rinsing with PBS, cells were fixed with Cytofix/Cytoperm (20 min, RT; BD Falcon). In order to label E- and N-cadherin, the method of cell detachment has been modified to maintain the connections between the cells. Control and treated cells were rinsed with PBS and peeled off using a solution of EDTA (4 °C, 10 min; Sigma-Aldrich). After centrifugation (8 min × 500× *g*), pellets were suspended in PBS with 2% FBS (Lonza Group, Ltd.) and stained for selected proteins. For the labeling of vimentin, the primary mouse anti-vimentin antibody (dilution 1:100, 1 h, RT; Sigma-Aldrich), for E-cadherin, the primary mouse anti-E-cadherin antibody (dilution 1:100, 1 h, RT; Sigma-Aldrich) and, for N-cadherin, mouse anti-N-cadherin antibody (dilution 1:100, 1 h, RT; Abcam) were used. In all cases, the secondary anti-mouse AlexaFluor 647 (dilution 1:500, 1 h, RT, dark; Invitrogen and Thermo Fisher Scientific, Inc.) was applied. In turn, for F-actin staining, AlexaFluor 488 conjugated with phalloidin (dilution 1:40 in PBS, 20 min, RT, dark; Invitrogen and Thermo Fisher Scientific, Inc.) was used. The measurement was performed using a Guava easyCyte 6HT-2L Benchtop Flow Cytometer (Merck KGaA) and FlowJo vX0.7 software (FlowJo LLC).

### 4.9. Statistical Analysis

The results obtained in all experiments were presented as the mean ± standard deviation (SD). Statistical analyses were performed using GraphPad Prism version 6.0 (GraphPad Software, Inc., La Jolla, CA, USA), and *p* < 0.05 was considered statistically significant. The compliance of the distribution of variables against the normal distribution was assessed using the Shapiro-Wilk test. In the case of the MTT assay, the Wilcoxon test was used, comparing the obtained data with the hypothetical value for the control group absorbance considered as 100%. For the ROS analysis, two-way ANOVA was used. To compare the differences between untreated and treated cells during the analysis of cell death, cell cycle and level of fluorescence intensity of the selected proteins, as well as wound healing, migration and invasion tests, the nonparametric Kruskal-Wallis with Dunn’s post hoc test was used.

## 5. Conclusions

In summary, we suggest that the treatment of A549 cells with a combination of PL and SAN in the ratio 4:1 indicates synergism without cytotoxic effects on normal lung cells. The interaction of the combined actions of the alkaloids depends on the type of cancer, which is probably associated with their individual response to the compounds. Furthermore, the action of the combination of the alkaloids is associated with an increase in the level of the ROS. Both PL and SAN have a cytotoxic effect and reduce the metastatic potential of A549 NSCLC cells. In turn, the combination of the alkaloids in the ratio 4:1 enhances this effect by promoting an inhibition in the cell migration and invasion, which are associated with the reorganization of F-actin and vimentin and a reduction in the expression level of EMT markers. The obtained results indicate a wide spectrum of PL and SAN activities, as well as the possibility of using their combination to increase the cytotoxic effects on the NSCLC A549 cell line.

## Figures and Tables

**Figure 1 molecules-25-03045-f001:**
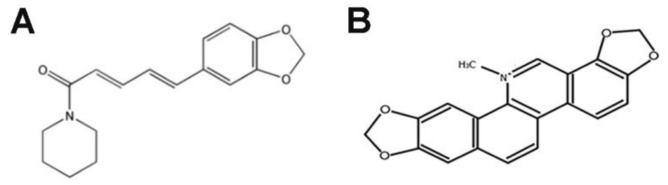
Chemical structures of piperlongumine (**A**) and sanguinarine (**B**) [12,26].

**Figure 2 molecules-25-03045-f002:**
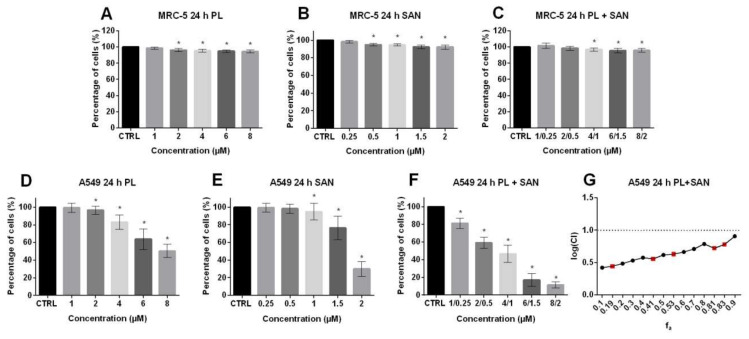
The cytotoxic effects of piperlongumine (PL) and sanguinarine (SAN) individually and in combined treatment on the cell viability of MRC-5 and A549 cells. The analysis was based on the results obtained from the MTT assay. Cells were treated for 24 h with PL at concentrations 1–8 µM (**A**,**D**); SAN at doses of 0.25, 0.5, 1, 1.5 and 2 µM (**B**,**E**) and their combination in ratio 4:1 (**C**,**F**). Data represent the mean values ± SD obtained from 6 independent replicates (*n* = 6). Statistically significant differences in comparison to untreated cells, where survival was estimated as 100%, were marked as * (*p* < 0.05; Wilcoxon test). (**G**) The combination index plot for the PL and SAN cotreatment in A549 cells in the range of fraction affected (f_a_) from 0.1 to 0.9. Combination index (CI) < 1—synergism, CI = 1—additive effect and CI > 1—antagonism. For real measuring points, the values have been marked in red.

**Figure 3 molecules-25-03045-f003:**
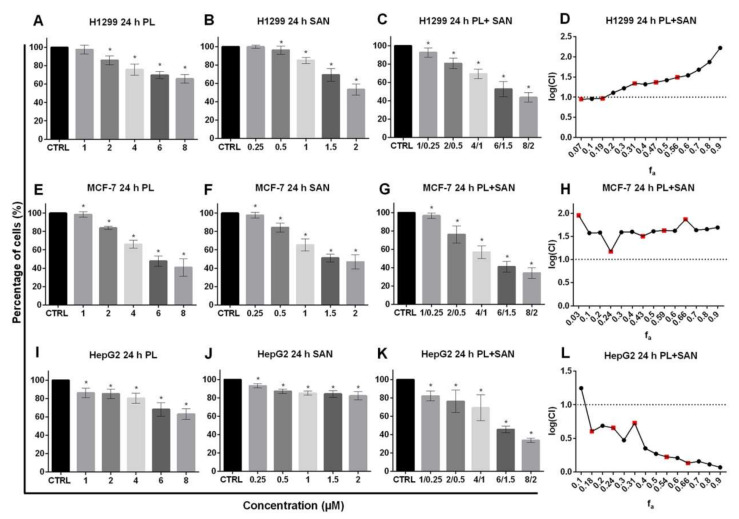
The cytotoxic effects of piperlongumine (PL) and sanguinarine (SAN) individually and in combined treatment on the cell viability of H1299, MCF-7 and HepG2. The analysis was based on the results obtained from the MTT assay. Cells were treated for 24 h with PL at concentrations 1–8 µM (**A**,**E**,**I**); SAN at doses of 0.25, 0.5, 1, 1.5 and 2 µM (**B**,**F**,**J**) and their combination in ratio 4:1 (**C**,**G**,**K**). Data represent the mean values ± SD obtained from 6 independent replicates (*n* = 6). Statistically significant differences with respect to untreated cells, where survival was estimated as 100%, were marked as * (*p* < 0.05; Wilcoxon test). The combination index plot for the PL and SAN cotreatment in H1299 cells (**D**), MCF-7 cells (**H**) and HepG2 (**L**) in the range of f_a_ from 0.1 to 0.9. CI < 1—synergism, CI = 1—additive effect and CI > 1—antagonism. For real measuring points, the values have been marked in red.

**Figure 4 molecules-25-03045-f004:**
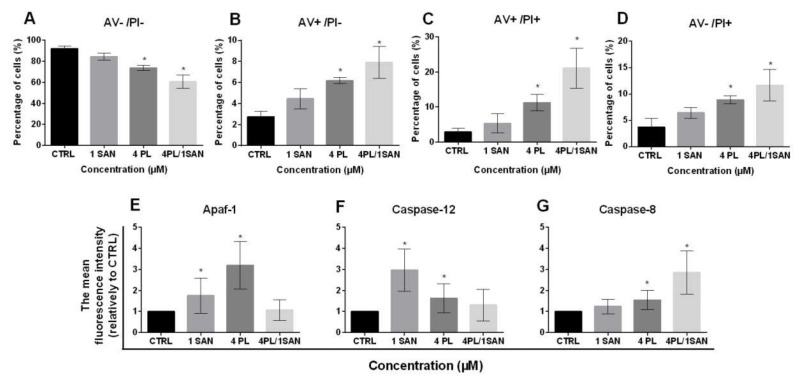
The effect of piperlongumine (PL) and sanguinarine (SAN) individually and in a combined treatment on the cell death of A549 cells. Annexin V/propidium iodide (AV/PI) assay. (**A**) The percentage of live cells (AV−/PI−). (**B**) The percentage of early-apoptotic cells (AV+/PI−). (**C**) The percentage of late-apoptotic cells (AV+/PI+). (**D**) The percentage of necrotic cells (AV−/PI+). Data represent the mean values ± SD obtained from 6 independent replicates (*n* = 6). The cytometric analysis of cell death-related proteins Apaf-1 (**E**), caspase-12 (**F**) and caspase-8 (**G**). Data represent the mean values ± SD obtained from 4 independent replicates (*n* = 4) and presented as folds in relation to the control (CTRL) (estimated as 1). Statistically significant differences in comparison to untreated cells were marked as * (*p* < 0.05; Kruskal-Wallis with Dunn’s post hoc test).

**Figure 5 molecules-25-03045-f005:**
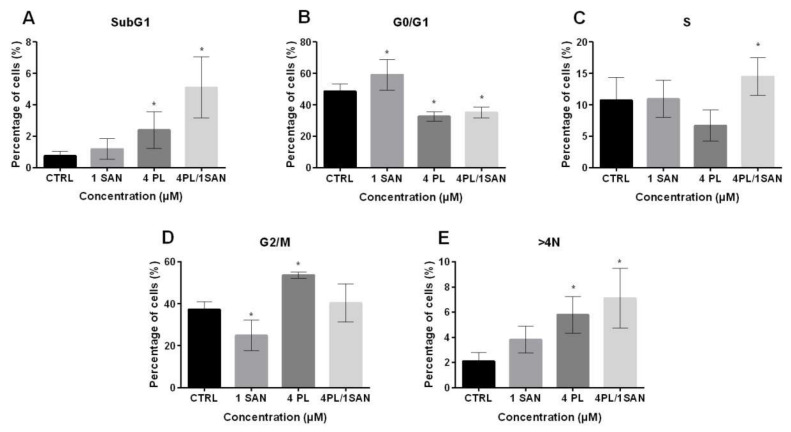
The effects of piperlongumine (PL) and sanguinarine (SAN) individually and in a combined treatment on the cell cycle of A549 cells. The percentage of cells in SubG1 phases (**A**), G0/G1 (**B**), S (**C**), G2/M (**D**) and >4N (**E**). Data represent the mean values ± SD obtained from 6 independent replicates (*n* = 6). Statistically significant differences in comparison to untreated cells were marked as * (*p* < 0.05; Kruskal-Wallis with Dunn’s post hoc test).

**Figure 6 molecules-25-03045-f006:**
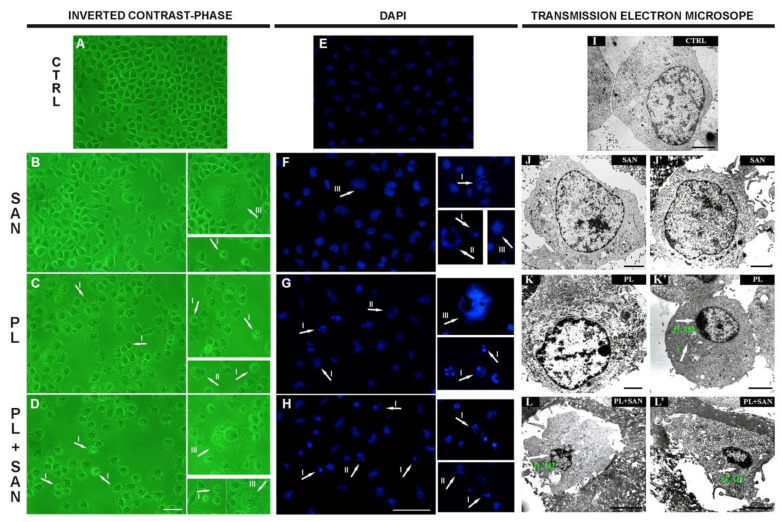
The effects of piperlongumine (PL) and sanguinarine (SAN) individually and in a combined treatment on the morphology, cell nuclei and ultrastructure of A549 cells. In the pictures (**A**–**H**), arrows indicate observed changes: contracted nuclei with a visible condensation of chromatin, apoptotic bodies (phenotype of apoptosis—**I**) and cells with features typical for mitotic catastrophe (multinucleated—**II** and giant cells—**III**). Bar—50 µm. In the pictures (**I**–**L**,**L′**), arrows mark visible changes: enlarged mitochondria (**I**), shrunken cell nucleus (**II**) and the condensation of chromatin (**III**). Bar—5 µm.

**Figure 7 molecules-25-03045-f007:**
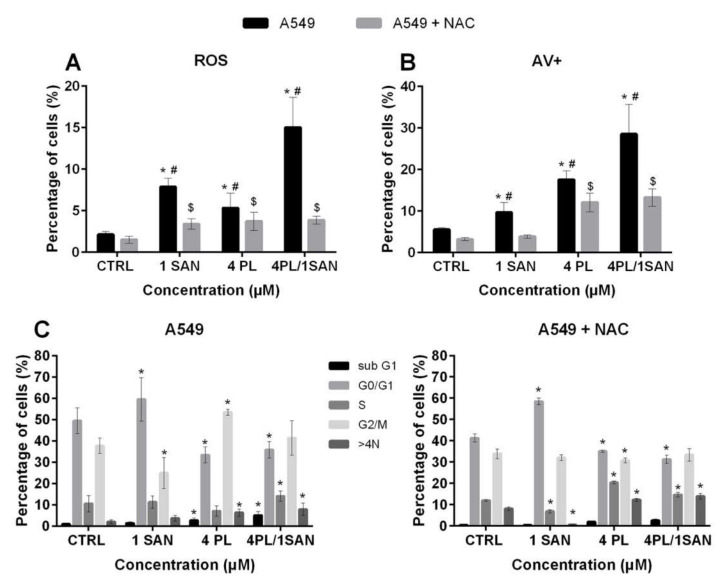
The synergistic effect of the alkaloids combination is related to the reactive oxygen species (ROS). Cells were treated for 24 h with PL at concentrations 1–8 µM; SAN at doses of 0.25, 0.5, 1, 1.5 and 2 µM and their combination in ratio 4:1. To inhibit ROS induction, A549 cells were preincubated for 2 h with 5-mM*N*-acetyl-l-cysteine (NAC) and then treated with alkaloids for 24 h. (**A**) The results obtained are presented as the percentage of ROS-positive cells relative to the sample size. (**B**) The comparison of results from the AV/PI assay for cells without (A549) and preincubated with NAC (A549 + NAC). Statistical significance was marked with * for the comparison between A549 cells without and with the preincubation with NAC, # for A549 cells without incubation with the ROS inhibitor in comparison to the CTRL and $ A549 cells with the addition NAC compared to untreated cells (*p* < 0.05; two-way ANOVA). (**C**) Cell cycle analysis for A549 and A549 + NAC. Data represent the mean ± SD obtained from 4 independent replicates (*n* = 4). Statistically significant differences in comparison to the untreated cells were marked as * (*p* < 0.05; two-way ANOVA).

**Figure 8 molecules-25-03045-f008:**
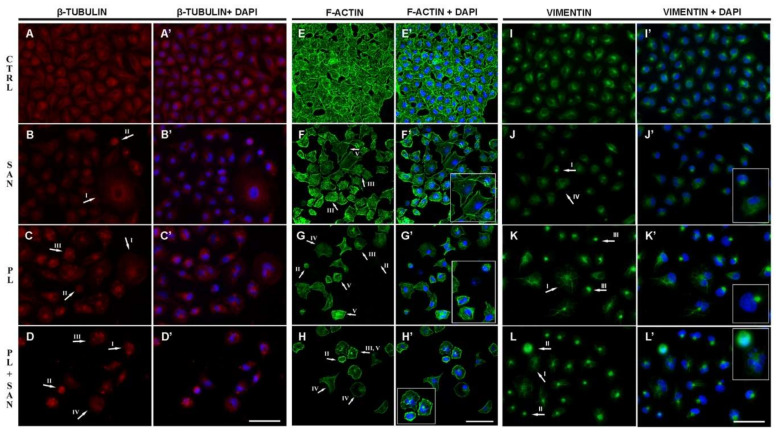
The effects of piperlongumine (PL) and the combination with sanguinarine (SAN) on cytoskeletal proteins in A549 cells: (**A**,**A′**–**D**,**D′**)—β-tubulin, (**E**,**E′**–**H**,**H′**)—F-actin, (**I**,**I′**–**K**,**K′**)—vimentin. Significant changes are marked with arrows—protein organization in the cells with the phenotype of mitotic catastrophe (**I**), apoptotic phenotype (**II**), point accumulation of the protein (**III**), dispersion of the protein’s network (**IV**) and stress fibers (**V**). Bar—50 µm.

**Figure 9 molecules-25-03045-f009:**
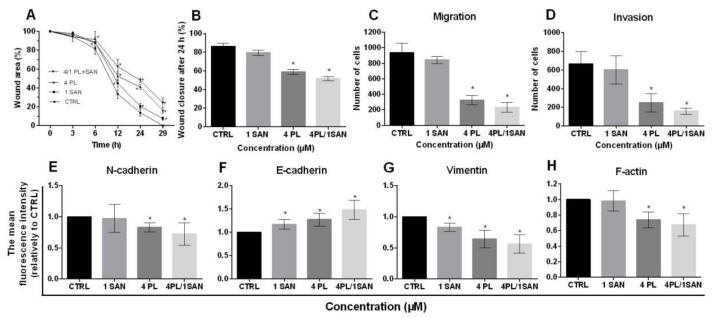
The effects of piperlongumine (PL) and sanguinarine (SAN) individually and in a combined treatment on the migration and invasion potentials of A549 cells. Wound-healing assay (**A**). The process of growing the surface “wound” after 0, 3, 6, 12, 24 and 29 h following the addition of 4-µM PL, 1-µM SAN and their combination in ratio 4:1. (**B**) The percentage of the surface of the wound area 24 h after mechanical scratching. Data represent the mean ± SD obtained from 3 independent replicates (*n* = 3). Statistically significant differences in comparison to the untreated cells were marked as * (*p* < 0.05; two-way ANOVA). Migration (**C**) and invasion assay (**D**). The average number of cells located on the outside of the insert. Data represent the mean ± SD obtained from 4 independent replicates (*n* = 4). Statistically significant differences in comparison to the untreated cells were marked as * (*p* < 0.05; Kruskal-Wallis with Dunn’s post hoc test). The cytometric analysis of the epithelial-to-mesenchymal transition (EMT)-related proteins N-cadherin (**E**), E-cadherin (**F**), vimentin (**G**) and F-actin (**H**). The results are presented as the fold of the results obtained for the CTRL (estimated as 1). Data represent the mean ± SD obtained from 4 independent replicates (*n* = 4). Statistically significant differences in comparison to untreated cells were marked as * (*p* < 0.05; Wilcoxon test).

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
