# Peer review of "The Synergistic Effect of Piperlongumine and Sanguinarine on the Non-Small Lung Cancer"

_molecules, 2020, doi:10.3390/molecules25133045_

Round 1
Reviewer 1 Report
This paper investigated the synergistic effect of piperlongumine and sanguinarine on the non-small lung cancer. It is of interest and can be accepted for publication after some revisions.
- The structures of the two alkaloids should be presented in a figure;
- Why you choose the two alkaloids for synergistic study? Do they have different mechanisms? These must be clearly shown in the introduction section.
- How do you determine the combination ratio of 4:1? The reason or background should be given.
- There may be many mechanisms for the synergistic effects on tumor cells. Why do you study the ROS and cytoskeletal proteins? Any backgrounds for these two alkaloids?
- Grammatic errors, e.g., Line 75, “analyzes”; line 110, “different”; etc.
Author Response
Dear Reviewer,
Thank you for all your valuable comments. All of the issues were carefully considered and the appropriate changes were introduced. They include: 1) The structures of piperlongumine and sanguinarine are presented in Figure 1 added in the Introduction section. 2) The whole manuscript was carefully rechecked and all of the mistakes (e.g., Line 75, “analyzes”; line 110, “different”; etc.) were corrected.
The other comments were also the basis for the improvement of the text:
1. Why you choose the two alkaloids for synergistic study? Do they have different mechanisms? These must be clearly shown in the introduction section.
The combination of piperlongumine and sanguinarine is innovative. However, the base were literature reports on the possibility of combining natural compounds for example curcumin, fisetin, or berberine [Zhang XZ et al. 2014; KlimaszewskaWiĹ›niewska et al. 2016, Ren et al. 2016; Wang et al. 2016]. The supportive arguments are selective ant-cancer properties and minimal effect on normal cells of both, piperlongumine and sanguinarine. On the other hand, similar mechanisms of action – the induction of apoptosis (but in different pathways) and reactive oxygen species can could be an advantage of the combination. Additional explanations were included: ď‚· in the Introduction section: “On the other hand, one of the strategies to improve anticancer treatment regimens may be to combine natural compounds such as alkaloids characterized by a wide spectrum of action. The validity of the assumptions is based on the literature reports on their selective action and minimal impact on normal cells. (..) “The developed combination of PL and SAN is innovative. However, similar mechanisms of action – the induction of apoptosis (but in different pathways) and reactive oxygen species were considered as a potential advantage. “ ď‚· in the Material and methods section: 4.1. Cell culture and treatment:” In the design of the experiment, MTT assay was carried out using a combinations of PL, SAN, and other alkaloids (aconitine, oxymatrine; data not shown). However, the most favorable interaction spectrum corresponding to synergism was estimated for the combination of PL and SAN.”
Zhang XZ, Wang L, Liu DW, Tang GY, Zhang HY. Synergistic inhibitory effect of berberine and d-limonene on human gastric carcinoma cell line MGC803. J Med Food. 2014;17(9):955-962. DOI:10.1089/jmf.2013.2967 Klimaszewska-Wisniewska A, Halas-Wisniewska M, Tadrowski T, Gagat M, Grzanka D, Grzanka A. Paclitaxel and the dietary flavonoid fisetin: a synergistic
combination that induces mitotic catastrophe and autophagic cell death in A549 nonsmall cell lung cancer cells. Cancer Cell Int. 2016;16:10. Published 2016 Feb 16. DOI:10.1186/s12935-016-0288-3 Ren K, Zhang W, Wu G, et al. Synergistic anticancer effects of galangin and berberine through apoptosis induction and proliferation inhibition in oesophageal carcinoma cells. Biomed Pharmacother. 2016;84:1748-1759. DOI:10.1016/j.biopha.2016.10.111 Wang K, Zhang C, Bao J, et al. Synergistic chemopreventive effects of curcumin and berberine on human breast cancer cells through induction of apoptosis and autophagic cell death. Sci Rep. 2016;6:26064. Published 2016 Jun 6. DOI:10.1038/srep26064
2. How do you determine the combination ratio of 4:1? The reason or background should be given.
The explanation was added in the section Material and methods: 4.1. Cell culture and treatment “The concentrations used in the studies were estimated experimentally, based on the available literature data and MTT assay.”
3. There may be many mechanisms for the synergistic effects on tumor cells. Why do you study the ROS and cytoskeletal proteins? Any backgrounds for these two alkaloids?
The aim of the study is based on literature reports which indicate the importance of ROS activation in the mechanism of action of both alkaloids. Furthermore, it is known that the cytoskeleton is important in basic cellular processes. There are few literature reports on the impact of PL and SAN on the organization of this structure. Therefore, this aspect has also been assessed in our studies. The next step will be expanding the research on the mechanism of action of PL and SAN combination in the context of other pathways, e.g. Wnt/β-catenin, PI3 K/Akt/mTOR using molecular biology methods.
Additionally, the explanation was added in the section Results: 2.5. Relationship between the synergistic effect of the alkaloids and ROS formation: “Literature reports indicate that the effect of PL and SAN is associated with ROS induction in various types of cells [17,28]. Thus, in this research on the synergistic action of the compounds induction of ROS was also determined”. Another information was included in the Discussion section: “Furthermore, available literature data indicate that action of PL and SAN may be associated with the formation of ROS in various types of cancers cells [17,28,61,62]. In our study, we also decided to assess whether the effect of a combination of the alkaloids is connected with the production of ROS”.
In the case of cytoskeleton analysis, the explanation was included in the Discussion section: “The results described above correspond to the observations regarding changes in morphology at the ultrastructural level and reorganization in main cytoskeleton proteins such as microtubules, vimentin, and F-actin. It is known that cytoskeleton is important in basic cellular processes. Furthermore, this is a particularly interesting aspect of the study as few reports describing the properties of selected alkaloids in the context of changes in the cytoskeleton are available.”
Best regards, Marta Hałas-Wiśniewska

Reviewer 2 Report
The authors report investigation of the influence of two natural alkaloids, piperlongumine and sanguinarine, and their mixture on the viability of cancer lung cells. The manuscript describes a very vast and in-depth investigation results proving the positive synergistic effect of a mixture of these two alkaloids. Therefore, the finding of the authors might be of interest for the researchers in the field of cancer treatment. However, some revisions regarding the text and English language are needed before the manuscript can be accepted for publishing.
- The first remark is more a suggestion. In my opinion, the chapters “Results” and “Discussion” could be merged into one “Results and Discussion” chapter as both of them contain a very valuable information, but it is not easy to correlate the results presented with the results published already by other scientists, when these are presented separately. It would be very helpful to get the information about the findings of other scientists on specific analysis and then immediately read the information about the results of the present study.
- In my opinion, the titles of subchapters 2.2, 2.3, 2.4, 2.5, and 2.7, instead of providing the conclusion of the investigation/analysis done, should indicate the investigation carried out in general, i.e. they could start with “Evaluation of the effect”, “Investigation of the influence”, or more simply “Effect of”, “Influence of”, etc.
- The quality of the Figures is very low. It is impossible to see the data presented. Colours could be used in the diagrams to make visualisation more facile.
- The authors should revise carefully a manuscript and organise the usage of abbreviations. There is no need to remind the meaning of abbreviations in each separate chapter. The common rule is that the full name and abbreviation in brackets is provided upon the first time it is used in the article and then just abbreviation is used.
- In considered to be not appropriate to write the scientific text in first person. All manuscript should be corrected to convert the sentences written in first person into passive voice.
- There should be a gap between number representing some value and corresponding measuring unit, i.e. 24h should be corrected as 24 h. All other similar cases should be corrected as well.
- In the References, volumes of the journals should be in italics.
- The suggested corrections for the English language or ambiguous places in the text are marked in the file attached.

Author Response
Dear Reviewer,
Thank you for all your valuable comments. All of the issues were carefully considered and the appropriate changes were introduced. The response to the comments and list of changes included is as follows:
1. The first remark is more a suggestion. In my opinion, the chapters “Results” and “Discussion” could be merged into one “Results and Discussion” chapter as both of them contain a very valuable information, but it is not easy to correlate the results presented with the results published already by other scientists, when these are presented separately. It would be very helpful to get the information about the findings of other scientists on specific analysis and then immediately read the information about the results of the present study.
Thank you for this suggestion. However, we think that combining these two large sections could create confusion, making the text more difficult to understand. The division into separate parts, Results, and Discussion is more common and most readers are used to it. Some researchers, to pre-check whether the article will be interesting for them, start reading from the Discussion, and then move on to the other parts. Therefore, we decided to stick to the current structure of the article.
2. In my opinion, the titles of subchapters 2.2, 2.3, 2.4, 2.5, and 2.7, instead of providing the conclusion of the investigation/analysis done, should indicate the investigation carried out in general, i.e. they could start with “Evaluation of the effect”, “Investigation of the influence”, or more simply “Effect of”, “Influence of”, etc.
All of the suggested corrections in the subchapters titles were introduced. The current titles are as follows: 2.2. Effect of PL, SAN and the combination of alkaloids on cell death in A549 cell line 2.3. Influence of PL, SAN, and their combination on cell cycle phases distribution 2.4. Impact of PL and SAN on alterations in morphology and ultrastructure of A549 cells 2.5. Relationship between the synergistic effect of the alkaloids and ROS formation 2.7. The influence of PL and its combination with SAN on migration and invasive potential of A549 cells.
3. The quality of the Figures is very low. It is impossible to see the data presented. Colours could be used in the diagrams to make visualisation more facile.
To improve the quality of the Figures, the font in all graphs has been enlarged and color in the diagrams has been changed from blue (which indeed was almost invisible) to red.
4. The authors should revise carefully a manuscript and organise the usage of abbreviations. There is no need to remind the meaning of abbreviations in each separate chapter. The common rule is that the full name and abbreviation in brackets is provided upon the first time it is used in the article and then just abbreviation is used.
The use of full names and abbreviations was corrected according to the suggestion.
5. In considered to be not appropriate to write the scientific text in first person. All manuscript should be corrected to convert the sentences written in first person into passive voice.
All sentences starting from e.g. “we observed” were converted into passive voice.
6. There should be a gap between number representing some value and corresponding measuring unit, i.e. 24h should be corrected as 24 h. All other similar cases should be corrected as well.
The whole manuscript was carefully rechecked and appropriate changes were introduced,
7. In the References, volumes of the journals should be in italics.
The volumes of the journals are now in italics.
8. The suggested corrections for the English language or ambiguous places in the text are marked in the file attached.
The manuscript was corrected according to the comments included in the attached files.
Best regards, Marta Hałas-Wiśniewska

Reviewer 3 Report
The manuscript "The synergisitc effect of piperlongumine and sanguinarine on the non-small lung cancer" by Halas-Wisniewska et al describes the citotoxicity and biological effects of the combination of two alkaloyds in non-small lung cancer cells. Overall, the data presented is convincing and is technically sound, and the conclusions are supported by the data presented.
However, a few general comments are outlined below:
- The authors should clarify clear what is the rational for the combination of the compounds or if it is solely on the previously described anti-cancer effects. It is only on the Discussion (line 460-462) that the authors address the rationale for the chosen ratio of the two compounds.
- Only one time point for the analysis are performed throughout the entire manuscript. I would suggest that more data could be included at leat for the MTT cytotoxicity assays.
- For the microscopic assays (Figs. 5 and 7), it is very difficult to observe the effects described in the text, in particular stainings with F-actin and vimentin. Close-up images must be provided to observe changes in detail. Images resolution must be improved as well.
Minor comments:
Line 35: replace testes by tests.
Line 444: replace repots by reports.
Author Response
Dear Reviewer,
Thank you for all your valuable comments. All of the issues were carefully considered and the appropriate changes were introduced. The response to the comments and list of changes is as follows:
- The authors should clarify clear what is the rational for the combination of the compounds or if it is solely on the previously described anti-cancer effects. It is only on the Discussion (line 460-462) that the authors address the rationale for the chosen ratio of the two compounds. The combination of piperlongumine and sanguinarine is innovative. However, the base were literature reports on the possibility of combining natural compounds for example curcumin, fisetin, or berberine [Zhang XZ et al. 2014; KlimaszewskaWiĹ›niewska et al. 2016, Ren et al. 2016; Wang et al. 2016]. The supportive arguments are selective anti-cancer properties and minimal effect on normal cells of both, piperlongumine and sanguinarine. On the other hand, similar mechanisms of action – the induction of apoptosis (but in different pathways) and reactive oxygen species can could be an advantage of the combination.
Additional explanations were included in the manuscript: ď‚· in the Introduction section: “On the other hand, one of the strategies to improve anticancer treatment regimens may be to combine natural compounds such as alkaloids characterized by a wide spectrum of action. The validity of the assumptions is based on the literature reports on their selective action and minimal impact on normal cells. (..) The combination of PL and SAN is innovative. Also, similar mechanisms of action – the induction of apoptosis (but in different pathways) and reactive oxygen species can be an advantage of the combination. “ ď‚· in the Material and methods section: 4.1. Cell culture and treatment:” In the design of the experiment, MTT assay was carried out using a combination of PL, SAN, and other alkaloids (aconitine, oxymatrine; data not shown). However, the most favorable interaction spectrum corresponding to synergism was estimated for the combination of PL and SAN.” and also “The concentrations used in the studies were estimated experimentally, based on the available literature data and MTT assay.”
Zhang XZ, Wang L, Liu DW, Tang GY, Zhang HY. Synergistic inhibitory effect of berberine and d-limonene on human gastric carcinoma cell line MGC803. J Med Food. 2014;17(9):955-962. DOI:10.1089/jmf.2013.2967 Klimaszewska-Wisniewska A, Halas-Wisniewska M, Tadrowski T, Gagat M, Grzanka D, Grzanka A. Paclitaxel and the dietary flavonoid fisetin: a synergistic combination that induces mitotic catastrophe and autophagic cell death in A549 non
small cell lung cancer cells. Cancer Cell Int. 2016;16:10. Published 2016 Feb 16. DOI:10.1186/s12935-016-0288-3 Ren K, Zhang W, Wu G, et al. Synergistic anticancer effects of galangin and berberine through apoptosis induction and proliferation inhibition in oesophageal carcinoma cells. Biomed Pharmacother. 2016;84:1748-1759. DOI:10.1016/j.biopha.2016.10.111 Wang K, Zhang C, Bao J, et al. Synergistic chemopreventive effects of curcumin and berberine on human breast cancer cells through induction of apoptosis and autophagic cell death. Sci Rep. 2016;6:26064. Published 2016 Jun 6. DOI:10.1038/srep26064
2. Only one time point for the analysis are performed throughout the entire manuscript. I would suggest that more data could be included at leat for the MTT cytotoxicity Assay
The MTT analysis were originally performed in three time points: 24h (included in the manuscript), 48h and 72h. Due to the large volume of the manuscript results of MTT assay after 48 h and 72 h treatment were added in the form of Supplementary material (Figure 1S). Appropriate reference to the Supplementary material was included in the Material and Methods: 4.2. Cell viability and type of drug interactions.
3. For the microscopic assays (Figs. 5 and 7), it is very difficult to observe the effects described in the text, in particular stainings with F-actin and vimentin. Close-up images must be provided to observe changes in detail. Images resolution must be improved as well.
To improve the quality of Figures, the font of the all graphs has been enlarged and colour in the diagrams has been changed from blue to red (Figure 6 - from white 9 to green). The appropriate inserts indicating changes described in the text were added. The sizes of images has been increased and the resolution was improved to 300 DPI.
4. Minor comments: Line 35: replace testes by tests. Line 444: replace repots by reports.
Errors have been corrected.
Best regards, Marta Hałas-Wiśniewska
